# NoxO1 Knockout Promotes Longevity in Mice

**DOI:** 10.3390/antiox9030226

**Published:** 2020-03-10

**Authors:** Tim Schader, Christina Reschke, Manuela Spaeth, Susanne Wienstroer, Szeka Wong, Katrin Schröder

**Affiliations:** Institute for Cardiovascular Physiology, Goethe-University, 60590 Frankfurt, Germany; Schader@vrc.uni-frankfurt.de (T.S.); Reschke@vrc.uni-frankfurt.de (C.R.); Spaeth@vrc.uni-frankfurt.de (M.S.); Wienstroer@vrc.uni-frankfurt.de (S.W.); SzeKaWong@gmx.net (S.W.)

**Keywords:** NADPH oxidase, NoxO1, longevity, mice

## Abstract

According to the free radical theory of aging, reactive oxygen species (ROS) have been proposed to be a major cause of aging for a long time. Meanwhile, it became clear that ROS have diverse functions in a healthy organism. They act as second messengers, and as transient inhibitors of phosphatases and others. In fact, their detrimental role is highly dependent on the context of their production. NADPH oxidases (Nox) have been discovered as a controllable source of ROS. NoxO1 enables constitutive ROS formation by Nox1 by acting as a constitutively active cytosolic subunit of the complex. We previously found that both Nox1 and NoxO1 were highly expressed in the colon, and that NoxO1-/- deficiency reduces colon health. We hypothesized that a healthy colon potentially contributes to longevity and NoxO1 deficiency would reduce lifetime, at least in mouse. In contrast, here we provide evidence that the knockout of NoxO1 results in an elongated life expectancy of mice. No better endothelial function, nor an improved expression of genes related to longevity, such as Sirt1, were found, and therefore may not serve as an explanation for a longer life in NoxO1 deficiency. Rather minor systemic differences, such as lower body weight occur. As a potential reason for longer life, we suggest better DNA repair capacity in NoxO1 deficient mice. Although final fatal DNA damage appears similar between wildtype and NoxO1 knockout animals, we identified less intermediate DNA damage in colon cells of NoxO1-/- mice, while the number of cells with intact DNA is elevated in NoxO1-/- colons. We conclude that NoxO1 deficiency prolongs lifetime of mice, which correlates with less intermediate and potentially fixable DNA damage at least in colon cells.

## 1. Introduction

Aging is the leading cause of death worldwide. Research has invested a lot of effort in prevention of aging, or at least of its consequences such as Alzheimers disease, stroke, muscle weakening, and others. So far, no cure from aging or age-related decline of health has been found. Besides the question of whether or not prevention of aging per se is a desirable goal, it is an eligible goal to age healthily and with fitness. Evidence-based studies indicate that longevity is based on two major factors; genetics and lifestyle choices. Molecular mechanisms of aging include proteostasis and quality control of proteins, DNA and RNA, as well as telomere length [1]. Genetic factors include SNPs (SNP—single nucleotide polymorphisms) and levels of gene expression, and thereby potentially epigenetic mechanisms [2]. 

Lifestyle in the context of aging often is associated with an altered formation of reactive oxygen species (ROS) [3]. Once just recognized as naturally occurring byproducts of mitochondrial respiration, today ROS are known to play essential roles for cellular homeostasis and as second messengers in many signaling pathways [4,5]. The family of NADPH oxidases (Nox) contributes to targeted and controlled ROS formation. NADPH oxidases can act constitutively active as in the case of Nox4, as Ca^2+^ dependent, Nox5 and the Duoxes1 and 2 or as acutely activatable complexes (Nox1-3), which require the cytosolic subunits p47phox and p67phox [6]. Both cytosolic subunits can be replaced by the activator NoxA1 and the constitutively active organizer NoxO1. Importantly, although NoxA1 and NoxO1 can be exchanged with p67phox and p47phox to activate Nox1-3 in vitro in an overexpression system, in vivo NoxO1 and p47phox are expressed in different cells and cannot substitute each other [7,8].

Recently we found a subunit of the Nox1 complex, namely NoxO1, in the colon to be involved in enterocyte differentiation [8]. In the colon, proliferation of partially differentiated epithelial cells was enhanced, while apoptosis was reduced in the absence of NoxO1 in murine colons, which leads to a reduced homeostasis and barrier function of the colon wall. Those data indicate that NoxO1 together with Nox1 mediates ROS formation and facilitates proliferation of colon epithelial cells. Thereby they contribute to gut homeostasis and enterocyte turnover. In NoxO1 knockout mice, ROS formation in the gut is reduced and the function of enterocytes may be disturbed. Eventually, NoxO1 knockout mice lose more weight and their colon was more prone to develop tumors in an AOM/DSS colon carcinoma model [8]. In humans, loss-of-function variants of Nox1 may not cause a disease, but represent context-specific modifiers that may worsen an inflammatory bowel disease [9]. We recently identified a role of human NOX1 in regulation of wound healing by altering epithelial cytoskeletal dynamics in pediatric-onset IBD patients with a rare sequence variant in NOX1. Importantly, loss of function of Nox1 is accompanied by a reduced ROS formation [10]. Klicken oder tippen Sie hier, um Text einzugeben. Assuming that the gut represents an organ system with dramatic impact on lifespan, we hypothesized a reduced lifespan for NoxO1-/- mice and analyzed whether or not the knockout of NoxO1 has any influence on lifespan in healthy mice. 

## 2. Materials and Methods

### 2.1. Animal Studies

We utilized NoxO1 knockout mice as previously described [11] and compared those to their wildtype littermates. NoxO1 mutant mice were obtained from the Jackson laboratory (Noxo1^hslt^, MGI:1919143) [12]. Mice were bred according to permit number 32.62.1 (Ordnungsamt Frankfurt). All mice were kept at the local facility and housed in the same breeding room with a 12/12 day and night cycle with chow and water ad libitum. Experiments conformed to the Guide for the Care and Use of Laboratory Animals published by the US National Institutes of Health and were approved by the local ethic committee and government. Mice were killed pain-free for organ/tissue harvesting and use for further in vitro investigations according to §4, Section 3 of the German animal protection law. Male mice were used for experiments. They were either housed for their lifetime or used for experiments at an age between 8–12 weeks. Weighting of the animals was performed as follows: Mice were placed in a box on a scale and the weight was written down. Femur length was measured once after preparation of the bone with the aid of a ruler. For the survival analysis we have to add a limitation statement, as the number of animals was quite low for a survival analysis. However, there was a statistically significant difference between the mouse strains even at these low numbers.

### 2.2. Organ Chamber Experiments

Organ chamber experiments were performed as described [13]. Aortas of the mice were dissected from surrounding tissue and rings of 1 mm length were cut and placed in the organ chambers. Relaxations to cumulatively increasing concentrations of acetylcholine (ACh) were recorded in vessels preconstricted to 80% of the maximal KCl (80 mmol/L)-induced contraction using phenylephrine in the presence of diclofenac (10 µmol/L). Relaxations are denoted as percent of the initial constriction obtained by phenylephrine. NO availability was estimated from the constrictor response to the NO synthase inhibitor N^ω^-nitro-L-arginine (L-NA, 300 µmol/L) in aortic rings preconstricted to 10% of the maximal KCl constriction using phenylephrine. 

### 2.3. Real-Time PCR

Organs (lung, colon, liver and kidney) were removed from 12-week-old mice, shock frozen in liquid nitrogen and mortared. RNA was isolated using RNA Miniprep Kit (Bio&Sell, Feucht, Germany), followed by DNase treatment (Promega, Walldorf, Germany). cDNA synthesis was carried out with SuperScript III Reverse Transcriptase (Invitrogen, Carlsbad, CA, USA) and oligo dT primers, semiquantitative real-time PCR was performed with ABsolute QPCR SYBR Green Mix and ROX as reference dye (Thermo Scientific) in an AriaMax cycler (Agilent Technologies) with appropriate primers. Relative expressions of target genes were normalized to eukaryotic translation elongation factor 2 (EF2), analyzed by delta-delta-Ct method and given as relative values compared to wildtype gut expression level. Primer sequences were as follows (Table 1).

### 2.4. DNA-damage Detection (Comet-Assay)

Guts were dissected into 1 mm pieces and subjected to a dispase digestion (Dispase II [Roche] 15 U/mL in modified Hanks-buffer w/o Ca^2+^ and Mg^2+^, 1 h at 37 °C). After digestion, debris were removed by filtering the suspension through a 70 µm sieve (Greiner, Bio-One, Frickenhausen). The resulting flow through was centrifuged at 500× *g* for 4 min. After removing the supernatant, cells in the pellet were resolved in 500 µL 0.5% BSA in PBS w/o Ca^2+^ and Mg^2+^. Cells were then counted, and 1 × 10^5^ cells were used for further analyzes in the comet assay. We utilized a kit from Trevigen (Wiesbaden, Germany) and followed the manufactures advice. Briefly: A suspension of 1 × 10^6^ cells/mL was mixed 1:10 with 5% low melting agarose and subjected onto slides coated with 1.5% normal melting agarose. Lysis of the cells was performed for 1 h at 4 °C in lysis buffer (2.5 M NaCl, 10 mM TRIS, 100 mM EDTA, pH = 10, 1% Triton X-100 and 10% SDS in double distilled water). Lysis was followed by a 20 min incubation of the slides on ice with the alkaline electrophoresis buffer (300 mM NaOH and 0.5 M EDTA). Subsequently electrophoresis was performed at 25 V for 20 min. Slides were washed three times with PBS and were stained with SYBR green. Pictures were taken with a confocal microscope LSM 510 Meta and quantification was done manually by three independent investigators determining the ratio of cell number/cells with comets, as described before [14].

### 2.5. Amplex Red Assay for H_2_O_2_ Formation

Colons were dissected, cleaned, minced and resuspended in Hepes-modified Tyrode’s solution containing Amplex Red reagent (50 μmol/L, Invitrogen) and horseradish peroxidase (2 U/mL, Sigma). The DPI-sensitive part was assessed by adding diphenylene iodonium chloride (10 μM, Sigma) to the reaction mixture. The reaction was carried out for 5 min and subsequently, fluorescence readings were made in triplicate in a 96-well plate at Ex/Em = 540/580 nm using 200 μL supernatant of each sample. Importantly the number of the repeats is low (n = 3), therefore the SEM is high.

### 2.6. Statistics

All values are mean ± SEM. For survival a Kaplan–Meyer curve-analysis was performed. Relaxations of the aortic rings were calculated from individual dose-response curves. Statistical analysis included the Shapiro–Wilk normal-distribution test, and were carried out by ANOVA for repeated measurements, followed by Fisher’s least significant difference test or *t*-test, if appropriate. Values of *p* < 0.05 were considered statistically significant

## 3. Results

Survival of male NoxO1 knockout mice was better than that of wildtype mice (Figure 1A). Postnatal growth rates were negatively correlated with adult lifespan in mammals [15]. Accordingly, we compared the growth rate of wildtype and NoxO1-/- mice. Although weight gain tends to be smaller in NoxO1-/- mice, the relative increase in weight in the first 5 months was not significantly different from that of the wildtype mice (Figure 1B) and no significant difference in femur length was observed at the age of 3 months (Figure 1C). In addition, NoxO1-mutant mice did not show significant differences in young age body weight (Appendix A). By appreciating the fact, that the animal numbers are relatively small, we aimed to avoid overlooking an effect of lower body weight, which appeared to be obvious: 

The most prominent reason for low weight gain in young animals is caloric restriction. Caloric restriction may develop from enterocyte dysfunction, as suggested by our previous study [8]. Alternatively, gut pain may be a reason for less food consumption. Accordingly, we analyzed indicators for caloric restriction such as sirtunin1 (Sirt1) and peroxisome proliferator-activated receptor gamma coactivator 1-alpha (PGC1α). Sirtuins can modulate oxidative metabolism as well as the biogenesis and turnover of mitochondria, with PGC1α being a key player therein [16]. Accordingly, we analyzed the expression of Sirt1, PGC1α. For this analysis, lung, liver, and kidney were chosen, as they represent organs exposed to toxins and involved in detoxifying the body. Further, all those organs, at least in the mouse have a self-repair capacity and therefore age-related deficits or mutation-collections may be more detrimental or better visible than in other organs. The colon was chosen, as it represents a side of major NoxO1 expression, which is exposed to potential toxic substances (food, bacteria, and their metabolic products) and is self-renewed on a regular basis. The results of the quantitative PCRs from different organs in Figure 2 show no difference on mRNA level of the two genes between wildtype and NoxO1 knockout mice. Importantly, mRNA does not necessarily reflect the activity of the protein. One other downstream effector of Sirt1 is the expression of endothelial nitric oxide synthase (eNOS) [17]. Evidence exists that eNOS activity is strongly associated with longevity [18]. We therefore analyzed endothelial function and NO formation in aortas from wildtype and NoxO1 knockout mice. Figure 3 shows no significant difference, neither in endothelial dependent relaxation nor in NO formation. We conclude that other mechanisms than those associated with caloric restriction may prolong the life of a NoxO1 knock out mouse.

Accumulation of DNA damage is a major reason for age-related effects. Two reasons led us to analyze DNA double strand breaks in isolated cells of the colon: (1) NoxO1 is mainly expressed in the colon; (2) is much more complex: DNA single strand breaks often occur due to hydrolysis and oxidative damage [19], while double strand breaks often are a consequence of exogenous sources. In the absence of exogenous stress, in the course of the cell cycle and thereby DNA replication, DNA is vulnerable to damage. If unrepaired, this can promote genomic instability. The majority of spontaneous DNA double strand breaks appear in the context of DNA replication [20]. Gut enterocytes represent a cell population with one of the highest turnover and replication rates in the body. We found more cells with intact DNA and fewer cells with intermediate comets in NoxO1-/- mice, while the relative number of full comets was similar (Figure 4A). Lower levels of DNA double strand breaks in cells of NoxO1-/- mice were supported by a trend of lower hydrogen peroxide formation in colon tissue of these mice (Appendix A). We speculate that at least some DNA repair mechanisms work better in NoxO1-/- cells than in wildtype cells. However, no difference in mRNA expression of the analyzed genes related to DNA repair was found (Figure 4B). Other mechanisms such as redox control of the involved proteins may apply. Alternatively, none-redox related binding partners of NoxO1 as suggested by the bio-informatics platform of protein–protein interaction BioGRID 3.5 may provide a future direction to identify the mechanism of better DNA repair in NoxO1 knock out (Figure 4C). Out of the 8 identified potential binding partners of NoxO1, 5 interact with the ATP-dependent DNA helicase Q4 (RECQL4), which is essentially involved in DNA double strand break repair [21]. To deeply analyze this hypothesis however, requires much more work and is far beyond the scope of the current manuscript. Nevertheless, extrapolation of an improved DNA repair in the gut to the whole body would represent a reason for improved lifespan of NoxO1-/- mice.

## 4. Discussion

This study indicates that the knockout of NoxO1 prolongs lifespan of mice. The most consistent factor promoting longevity is dietary restriction or in other words a low body weight [22]. One effect of low body weight might be the expression and activity of Sirt1 and PGC1α, which are upregulated in response to caloric restriction [23]. However, neither Sirt1 nor PGC1α was upregulated in NoxO1 deficient organs. Accordingly, the Sirt1 downstream target eNOS was not differentially activated in NoxO1-/- mice. We conclude that Sirt1 is not the preliminary effector to prolong the lifespan of NoxO1-/- mice, when compared to wildtypes. Although not significantly different from wildtype mice, low body weight in NoxO1 mice may have other beneficial effects. Indeed, in rhesus monkeys, late onset (from the age of 16, with maximum age of 43) caloric restriction improves health and potentially survival. Importantly, in monkeys improved longevity is most likely the result of less incidence of diseases such as cancer and insulin resistance, rather than being directly related to caloric restriction or low body weight [24]. In fact, despite several studies in lower organisms, such as worms and flies, supporting the assumption of improved longevity by caloric restriction, it cannot be applied to humans. Actually, white men live longest, when slightly overweight (around 26 kg/m^2^) and women survive better in the upper range of normal BMI (around 23 kg/m^2^) [25,26]. A speculative explanation for why there is such an enormous discrepancy between worm and human or monkey, could be contentedness. In humans eating and the resulting saturation are major aspects of life quality, and happy persons live longer. At least high and medium lifetime life satisfaction promotes longevity over low lifetime life satisfaction [27]. However, the authors fear that lifetime life satisfaction does not apply for laboratory animals, such as mice. 

Another obvious thought, facing a prolonged lifespan in a NoxO1 deficient mouse, would be that this is another proof of the free radical theory of aging [28]. This theory was later more specified to be the theory of aging due to accumulation of damage by mitochondrial ROS [3]. Indeed, deficiency of another NADPH oxidase, Nox4 has no impact on lifetime in mice, although Nox4 is expressed throughout the body and constitutively produces H_2_O_2_ [29]. Nevertheless, ROS-mediated DNA damage increases with age and can be reduced by caloric restriction in mice [30]. We analyzed the basal level of DNA double strand breaks in a comet assay. According to the assumption that knock out of NoxO1 reduces ROS formation and subsequent DNA damage, fewer double strand breaks were observed in NoxO1 deficient mice. Although this result fits the expectations one should remember that the pure comet assay indicates DNA double strand breaks only, but not DNA oxidation. Such an approach would require analyses using for example formamidopyrimidine-DNA glycosylase, which specifically recognizes oxidized purines and endonuclease III, which specifically recognizes oxidized pyrimidines [30]. As mentioned above, DNA double strand breaks occur in the course of cell division and proliferation. Other reasons of DNA double strand breaks may arise in the course of transcription. R-loops, three-strand nucleic acid structures, RNA:DNA hybrids, have been linked to DNA damage [31]. DNA double strand breaks may disappear, if DNA-de-novo synthesis or DNA excision-repair is enhanced. Indeed, capability of DNA de-novo synthesis in isolated fibroblasts can be correlated to lifespan [32]. This correlation however may not be the only reason for longer lifespan. It rather appears that DNA excision-repair represents an advantage, if organisms are challenged, instead of directly interfering with aging [32]. The lower level of intermediate comets in NoxO1 mice indicate a more efficient repair of damaged DNA. Coordinated repair of the DNA lesions is a prerequisite for the cells to survive and return to their original state. In contrast, irreparable lesions trigger a persistent DNA damage response which can then initiate cellular senescence or apoptosis [33,34]. In line with that, NoxO1-/- mice have less apoptotic enterocytes and develop DSS/AOM-induced colon cancer easier than wildtype mice [8].

In the light of the results of this study, fewer DNA double strand breaks in colon cells and prolonged lifespan, free radicals, if at all, should be rather considered as part of a network of events leading to aging than being its major cause. Other mechanisms that impact aging include metabolic stability of regulatory networks and reduction of cellular repair mechanisms. This study suggests that at least in mice, the knockout of NoxO1 extents lifespan, potentially by improving DNA repair mechanisms.

## Figures and Tables

**Figure 1 antioxidants-09-00226-f001:**
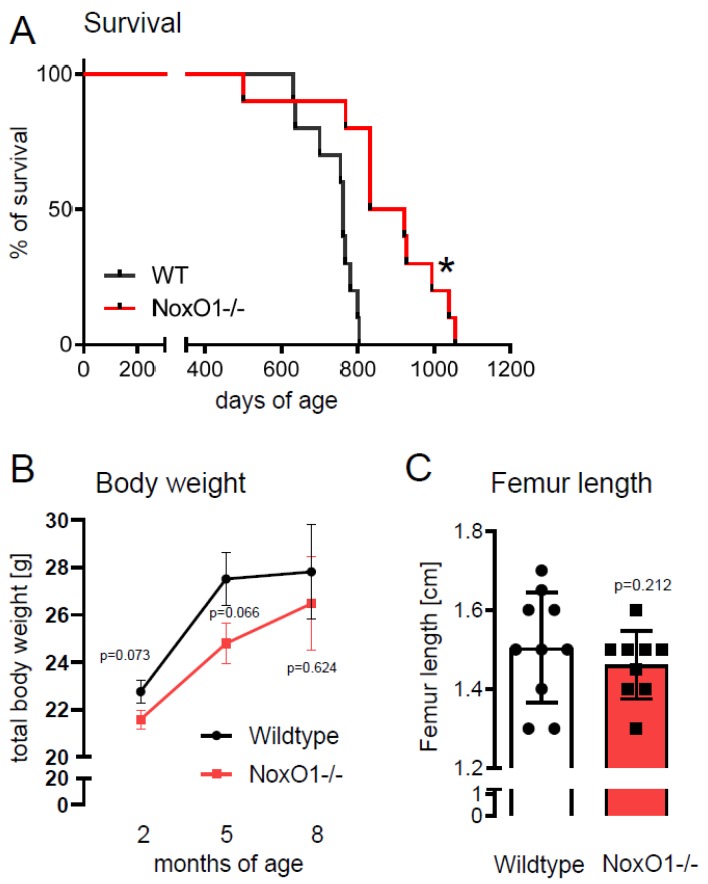
(**A**) Kaplan–Meyer curve for survival; (**B**) body weight development; and (**C**) femur length at the age of 12 weeks of male wildtype and NoxO1-/- mice. n = 9–10; * *p* < 0.05.

**Figure 2 antioxidants-09-00226-f002:**
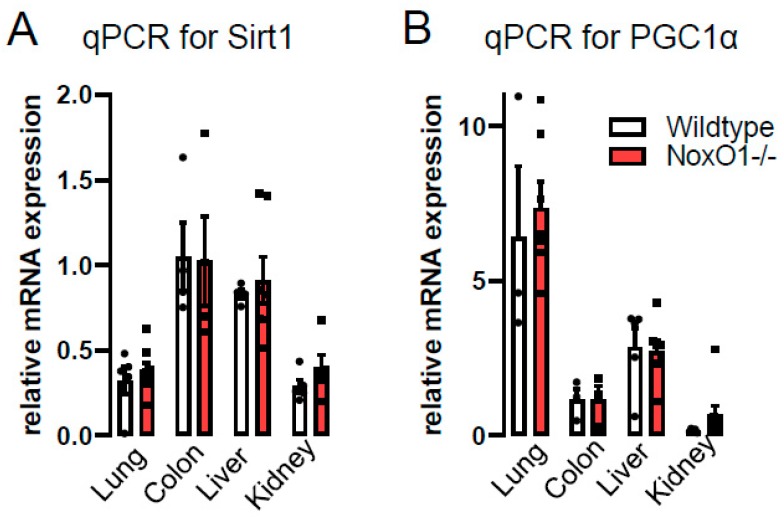
mRNA expression of (**A**) sirtuin1 and (**B**) PGC1α in the organs indicated. n = 5 (5 male mice at the age of 12 weeks with 4 quantitative PCRs for each organ).

**Figure 3 antioxidants-09-00226-f003:**
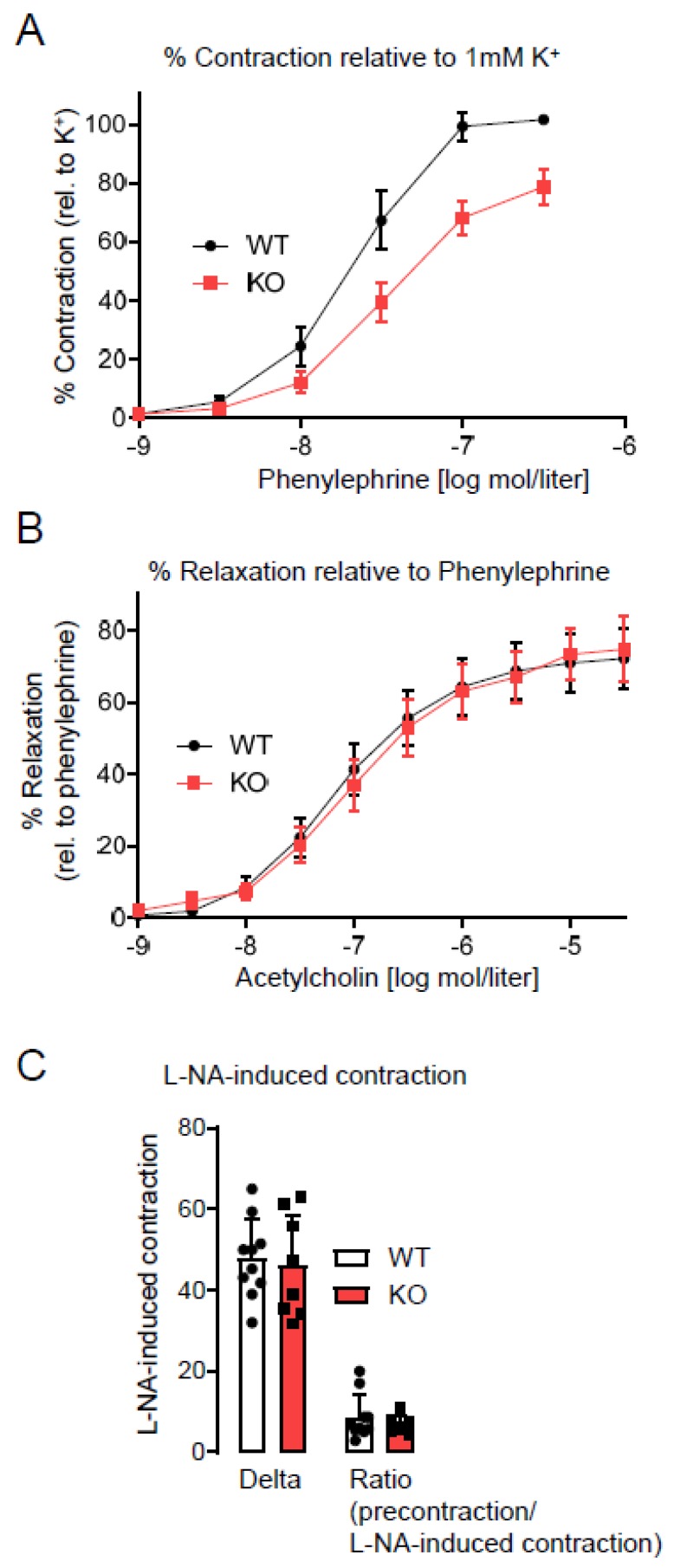
Vessel function with (**A**) contraction in response to phenylephrine and (**B**) relaxation in response to acetylcholine at the concentrations indicated. (**C**) L-NA induced contraction as measure of NO formation. n = 10; male mice at the age of 12 weeks.

**Figure 4 antioxidants-09-00226-f004:**
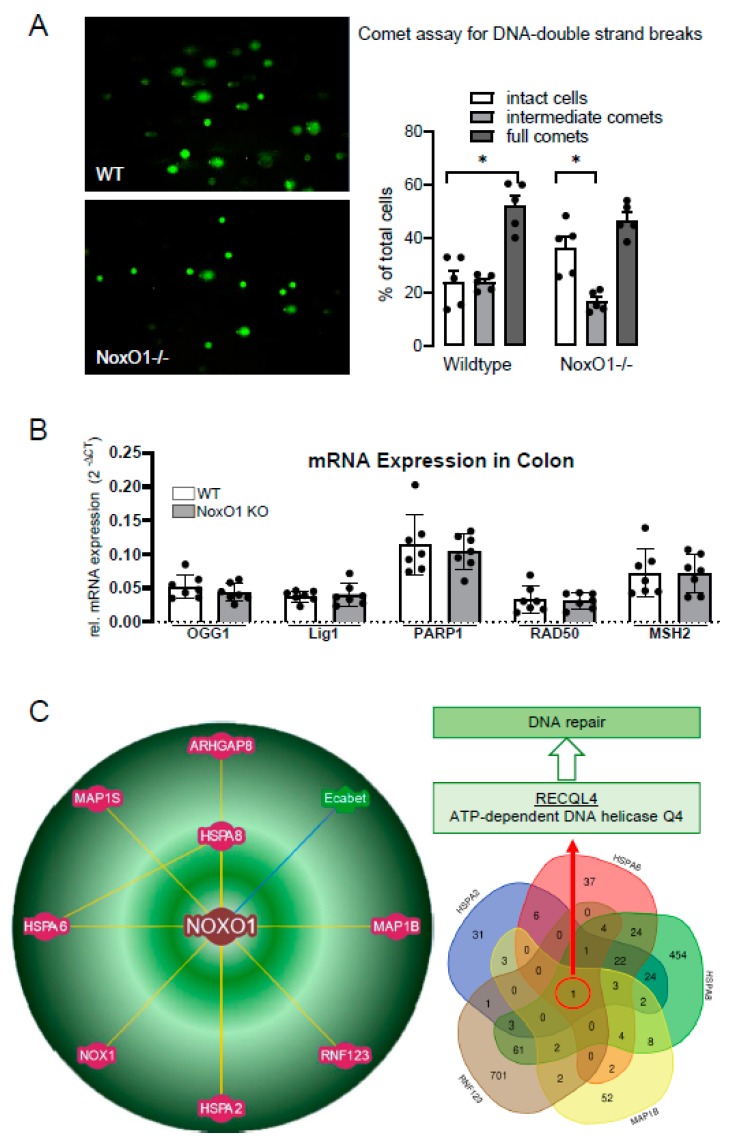
(**A**) Comet assay of isolated enterocytes of male mice at the age of 12 weeks with representative pictures and statistics. n = 5 (5 mice with 5 pictures for individual means); (**B**) QPCR for the genes indicated in colon tissue of 7 mice; * *p* < 0.05. (**C**) BioGRID 3.5 based network of potential binding partners of NoxO1 and Venn diagram of secondary interaction partners (VIB/UGent; Bioinformatics and Systems Biology; Gent; BELGIUM).

**Table 1 antioxidants-09-00226-t001:** Primer sequences.

Gene.	Forward/Reverse	5′-->3′
mOGG1	fw	GGAGCTGGAAACCCTACACAA
	rev	GGGTCTTGTCTCAGCAGTCT
mLig1	fw	AGAGCTGGGTGTTGGTGATG
	rev	TCCCCCTTCTCAGCTACCTC
mPARP1	fw	GCGGAGAAGACATTGGGTGA
	rev	ACCATCTTCTTGGACAGGCG
mRAD50	fw	TCCCTCCTGGAACCAAAGGA
	rev	TCGAAACTGCAGGCGAATCT
mMSH2	fw	GGCCCAGGATGCCATTGTTA
	rev	AAGTGAGCCAGCACATCGTT
mSIRT1	fw	ATGCTGGCCTAATAGACTTG
	rev	AGCACCGTGGAATATGTAAC
mPGC1α	fw	ACAGCTTTCTGGGTGGATTG
	rev	TGTCTCTGTGAGAACCGCTA
hmr-EF2	fw	GACATCACCAAGGGTGTGCAG
	rev	GCGGTCAGCACACTGGCATA

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
