# Peer review of "NoxO1 Knockout Promotes Longevity in Mice"

_antioxidants, 2020, doi:10.3390/antiox9030226_

Round 1

Reviewer 1 Report

Schader et al describes the elongation of life span in mice by NoxO1 KO. They identified that NoxO1- KO mice shows less intermediate damage in colon cells. Based on this finding, they speculated that NoxO1-KO provide with better DNA repair capacity. Following points shall be addressed properly before further process. In addition, English writing shall be re-edited to be much clearer.

Did the authors check food consumption? Did the authors the antioxidant status of the gut tissue? Causality between DNA damage and longevity is not fully demonstrated. Further evidence shall be necessary. What is the number of mice of Fig.1A? If n=9, the statistical power is too low for the survival analysis. What was the cause of deaths? As explained in introduction, NoxO1-KO may increase tumorigenesis. Is there any data to support that the quality of life of NoxO1-KO is similar to wild type? 3A suggests that contractile response to phenylephrine was lower in KO mice. This data shall be discussed further. Low blood pressure may be helpful for longevity. Is the statistics used in Fig.4 correct? Comparator shall be wild type?

Author Response

Please find attached a point by point reply as pdf.

Reviewer 2 Report

NoxO1 knock out promotes longevity in mice.

The authors propose that NoxO1 deficiency prolongs the life of mice by improving DNA damage repair.  The current paper is an extension of previous work using the same knockout model.  The experiments included here present some interesting and potentially novel findings, however, the authors are a little premature in describing enhanced DNA damage repair as the mechanism involved in promoting longevity in mice.

Specific comments:

[1] Abstract – the authors should provide additional evidence that the cells are healthier.

[2] Abstract – the final sentence shows an association rather than direct evidence – this should be amended. 

[3] Introduction – all statements should be clarified with a reference for example, “Lifestyle in the context of ageing ……. ROS”. 

[4] Methods – in the DNA-damage detection section the cells are not mentioned or how they were extracted.

[5] Figure 4 – the cell type is not mentioned in the legend.

Author Response

(The authors gave the same response as above.)

Reviewer 3 Report

The manuscript entitled "NoxO1 knock out promotes longevity in mice " has a special relevance in the field of longevity and oxidative stress, and It is of great interest to our journal.
To increase the quality of the manuscript, please consider some of the following recommendations for publication.

In the article you do not describe who as this fact the transgenic animal "We utilized NoxO1 knock out mice as previously described [5]" Specify the origin, the reference is not concrete.

You must report on the permission data of the relevant ethics committee. The age and sex of animals, etc. The age of animals, organs and treatment used should be more specific.

You should check that in the study tissues, NOX activity is understated. You can check by PCR for the drop in gene expression. The methodology is not specified.

How has the length of the femur been measured? Specify in methods.
The experiments in Figure 4 are not described in the methodology.
You must add some conclusions. These appear to be related to colon cells. Do you think that is the cause of the increase in longevity?
Other suggestions. To confirm the increase in longevity, measures should be made to animals related to healthy aging. Some such as, glucose tolerance, grip strength, hair growth, copulatory activity, oxidative stress parameters (MDA, carbonylated prot., GSH/GSSH), etc.
Whether in the discussion and conclusion you indicate at the end of the results you want to demonstrate greater effective in DNA repair, you must confirm the mechanism, I suggest you measuring by PCR enzymes involved it.
If you confirm this hypothesis the article will gain quality and will be proposed for publication.

The research is good.

Add conclusions as a final section.

Author Response

(The authors gave the same response as above.)

Reviewer 4 Report

Reviewer comments

Abstract

The background part in the abstract is very extensive while the central part of the work, experimental procedure and results are very schematic. It would be convenient to provide more information and more detail on these two sections in the abstract.

Line 9, correct: “Meanwhile, it became…”.

Introduction

In the abstract many sentences need some references to support the statement. For example, lines 29, 32, 34, 35 or 36.

All abbreviations should be described the first time used. For example: SNP - single nucleotide polymorphisms. The entire manuscript should be revised in accordance.

When introducing NADPH oxidases, more information about the general characteristics, distribution and physiological function should be provided to better focus the further research.

Materials and methods

In general, the experimental procedure is very poor since there are a lot of information lacking making complex to understand all the procedures or replicate the experiments if required

A brief explanation about NoxO1 knock should be added in addition to the cite reference. Also, information about the age, weight, number of animals used or from where the animals were purchased are needed.

The procedure to evaluate the longevity of the animals, nor the variables to be considered, the appearance of pathologies, etc., is NOT explained at all.

How the animals were weighed and how femur length was measured and how often?

Organ Chamber Experiments should also be better described. How the organs were processed, which concentrations of acetylcholine and phenylephrine were used, etc.

For the PCR analysis, at what time were the tissues obtained? And which tissues were dissected and why?

How the suspension of cells was obtained for the DNA-damage detection? How the comets were analysed to determine the degree of damage?

Was the normality of the results obtained analysed to further use a parametric analysis such as ANOVA? The Kaplan Meyer curve analysis was not indicated in the section.

There is no reference to the ethical aspects of the study, something obligatory in this type of studies.

Results

In the results section it should be avoid any discussion or introductory information of the obtained data. In this sense the explanation about the effect of lower body weight should be removed from this section.

The phrases of lines 140-142 are speculative since no repair mechanism has been analysed so it must be nuanced or eliminated.

Discussion

Low body weight is associated with reduced incidence of many cardiometabolic diseases which can contribute to better life expectancy. Did the authors determine any parameters related to obesity such as lipid profile or insulin resistance? This fact would be interesting to know if the less weight has an impact on longevity. Anyway, this possibility should be considered in the discussion.

The explanation about “Nox4 has no impact on life time in mice” is not clear, and should be better described.

Author Response

(The authors gave the same response as above.)

Reviewer 5 Report

Antioxidants-698405

NoxO1 knock out promotes longevity in mice

Several comments for this manuscript:

The data provided is slightly inadequate “NoxO1 knock out promotes longevity in mice”. There should be more data evidence. Materials and Methods should be description detailed, especially in “Animal studies” section.

Author Response

(The authors gave the same response as above.)

Round 2

Reviewer 1 Report

The reviewer sees that the authors made sufficient efforts. Still there are some points unclear that needs to be improved but they seem not possible in the present paper. Now it is acceptable for further processes.

Author Response

We thank the reviewer for this valuable comment and included the figure on DNA repair related genes into the manuscript as new Figure 4B.